# Reduction in organ–organ friction is critical for corolla elongation in morning glory

Ayaka Shimoki[1,10], Satoru Tsugawa [2,10], Keiichiro Ohashi[1], Masahito Toda[1], Akiteru Maeno [3], Tomoaki Sakamoto [4], Seisuke Kimura [4,5], Takashi Nobusawa[6], Mika Nagao[1], Eiji Nitasaka[7], Taku Demura [2], Kiyotaka Okada[8] & Seiji Takeda [1,9✉]

In complex structures such as flowers, organ–organ interactions are critical for morphogenesis. The corolla plays a central role in attracting pollinators: thus, its proper development is important in nature, agriculture, and horticulture. Although the intraorgan mechanism of corolla development has been studied, the importance of organ–organ interactions during development remains unknown. Here, using corolla mutants of morning glory described approximately 200 years ago, we show that glandular secretory trichomes (GSTs) regulate floral organ interactions needed for corolla morphogenesis. Defects in GST development in perianth organs result in folding of the corolla tube, and release of mechanical stress by sepal removal restores corolla elongation. Computational modeling shows that the folding occurs because of buckling caused by mechanical stress from friction at the distal side of the corolla. Our results suggest a novel function of GSTs in regulating the physical interaction of floral organs for macroscopic morphogenesis of the corolla.

[1] Graduate School of Life and Environmental Sciences, Kyoto Prefectural University, Kyoto, Japan. [2] Graduate School of Science and Technology, Nara Institute of Science and Technology, Ikoma Nara, Japan. [3] Plant Resource Development, Division of Genetic Resource Centre, National Institute of Genetics, Shizuoka, Japan. [4] Centre for Plant Sciences, Kyoto Sangyo University, Kyoto, Japan. [5] Department of Industrial Life Sciences, Faculty of Life Sciences, Kyoto Sangyo University, Kyoto, Japan. [6] Graduate School of Integrated Sciences for Life, Hiroshima University, Hiroshima, Japan. [7] Graduate School of Science, Kyushu University, Fukuoka, Japan. [8] Faculty of Agriculture, Ryukoku University, Shiga, Japan. [9] Biotechnology Research Department, Kyoto Prefectural Agriculture Forestry and Fisheries Technology Centre, Kyoto, Japan. [10] These authors contributed equally: Ayaka Shimoki, Satoru Tsugawa. ✉email: seijitakeda@kpu.ac.jp

Multicellular organ morphogenesis is fundamental for the development of structures and execution of functions in organisms. In plants, organ patterning is largely established by position-dependent cell differentiation rather than by cell lineages[1,2]. Positional information is determined by several mechanisms, including auxin-derived gradients, non-cell-autonomous action of transcription factors, and cell–cell communication mediated by ligands and receptors[2]. In addition to these molecular factors, mechanical regulation at the intra- and/or inter-organ level may affect organ development, especially in complex structures such as flowers.

Flowers are reproductive places in flowering plants and are generated by four distinct organs: two perianth organs (sterile floral leaves; the calyx and corolla) and two reproductive organs (fertile floral leaves; the androecium and gynoecium). The corolla plays a central role in attracting pollinators via their shape, color, and odorants: thus, corolla morphogenesis is important in natural pollination, as well as for agriculture and horticulture. Genetics-based analyses in *Arabidopsis thaliana* and *Antirrhinum majus* have shown that floral organ identity occurs in a concentric pattern by the combined functions of floral homeotic genes[3–5]. In addition to this basic mechanism, regulation of the growth direction of cells along the proximodistal axis is important for corolla patterning, which was shown by analyses of cell division patterns, clonal analysis, and computational modeling in *Antirrhinum majus*[6,7]. The complex 3D petal shape in orchids, such as twisting, helical twisting, saddle bending, or edge waving, can be reproduced by a computational model and actual construction with hydrogel[8], suggesting that physical strength derived from growth strain gives rise to plant organ structure. Although these analyses clearly explain plant organogenesis at the intra-organ level, little work has been done from a mechanical perspective on floral organ interactions during development.

To examine floral organ development, we focused on Japanese morning glory (*Ipomoea nil* (L.) Roth, syn. *Pharbitis nil* (L.) Choisy), a member of the Convolvulaceae family, whose flowers generate a trumpet-shaped corolla consisting of five fused petals (Fig. 1a, b). Japanese morning glory was originally introduced from China and Korea to Japan in the 8th century for use as a laxative, and breeders eventually began developing flowers with various shapes for ornamental purposes in the 17th century. Currently, more than 1500 lines are maintained at Kyushu University as part of the National BioResource Project (NBRP) in Japan. The wide variation in color and morphology seems to be generated by transposable elements, and several genes regulating flower color or morphology have been identified in *I. nil* and *I. purpurea*, a relative species of *I. nil*, by transposon tagging strategies[9–13]. The abundance of mutants of morning glory, together with genome sequence information[14], makes this plant a good model to investigate a novel mechanism of floral organ development.

Among the mutant lines, we analyzed 'cup flower' mutants with an appealing morphological trait—a folded corolla (Fig. 1c, d). This characteristic folding was described as a 'cup tube' ~200 years ago in Japan[15–17] and became one of the major phenotypes with high ornamental value in the early 20th century[18]: for example, a representative mutant line carries a blown-up corolla due to the combination of a cup tube and homeotic conversion of reproductive organs to a corolla (Supplementary Fig. 1). Despite the historical description and high ornamental value, the mechanism of corolla folding remains elusive.

Here, we describe the cellular and mechanical regulation of corolla folding in cup flower mutants. We found that glandular secretory trichomes (GSTs) on peripheral organs play an essential role in corolla elongation by reducing the friction between floral organs. Computational modeling consistently indicated that mechanical force arising from defects in GSTs was the primary cause of corolla folding in the cup flower lines. Our results suggest that proper corolla morphology depends on an active mechanism that reduces the friction between floral organs, proposing a critical mechanism to control mechanical conflict during flower development.

## Results

**Corolla folding in cup flower mutants.** The corolla of morning glory flowers consists of a basal tube and apical lobe generated by five fused petals. The tube part of the corolla in wild-type flowers (Tokyo Kokei Standard, TKS) elongates in a straight manner (Fig. 1a, b), whereas the tube of the mutant flower folds twice, generating a cylinder-like structure at the center of the flower (Fig. 1c, d). This phenotype is linked to the genetic locus *crepe* (*cp*) or *cp-reversed* (*cp-r*)[16], and nine lines and one line with mutations in *I. nil* and *I. purpurea*, respectively, exhibit a folded-corolla phenotype (Supplementary Fig. 2 and Supplementary Table 1). The *cp* locus gives rise to wrinkled leaf morphology, but leaves of *cp-r* are not wrinkled or are wrinkled little (Supplementary Fig. 3a), suggesting that corolla folding and wrinkled leaves are caused by different genes. F1 plants generated by a cross between *cp* and TKS bore flowers with a straight corolla (Supplementary Fig. 3b), and F2 plants from those self-crossed F1 plants displayed flowers with straight and folded corollas, with the segregation ratio of 33:8 (4.125:1, $P = 0.4171$ ($P > 0.05$) for 3:1 ratio, according to a chi-square test), suggesting that corolla folding is a recessive trait. The cross between *cp* and *cp* or *cp-r* resulted in folded-corolla plants in the F1 generation (Supplementary Fig. 3c, d), suggesting that the corolla defect was caused by the same gene in cup flower mutants.

Cup flower lines display flowers with different degrees of corolla folding. Some flowers appear normal, with a straight tube, and others are classified as 'traced', where the tube part shows some wrinkled regions (Fig. 1e), 'half', which are half-bent flowers (Fig. 1f), and 'cup', where the tube part folds completely and generates a cylinder-like structure at the center of the flower (Fig. 1d). Counting the number of flowers in the cup flower lines revealed that the ratio of the degree of corolla folding varied among lines (Fig. 1g).

In the developing floral buds of wild type morning glory, the corolla covered the primordia of the stamens and gynoecium (Fig. 1h and Supplementary Fig. 4a). At the same stage in the mutant flower, a small cavity appeared on the surface of the corolla (Fig. 1i). The cavity spread around the floral buds, circled the buds, and became deeper during bud development, resulting in the generation of a folded corolla (Supplementary Fig. 4b). The connection of the vascular bundle did not break at the cavity (Fig. 1j and Supplementary Movie 1), and the lengths of the tube and lobe were slightly shorter or were no different from those of the wild-type flowers (Supplementary Fig. 4c–e), suggesting that intrinsic corolla development was not affected in the mutants.

**Mechanical stress caused corolla buckling.** When the sepals were removed from the young floral buds of the mutant, straight elongation of the tube part was restored (Fig. 1k, l), suggesting that physical friction between the sepals and corolla caused folding. We tested this hypothesis through computational modeling using the finite element method based on the acropetal growth inflated by turgor pressure with friction of the perianth organ at the distal part of one side of the corolla. Compared with the case without friction (Fig. 2a and Supplementary Movie 2), the simulated corolla with friction bends at the region under the frictional area, which resembles a buckling phenomenon (Fig. 2b, c, Supplementary Movies 3 and 4)[19,20]. More friction generated

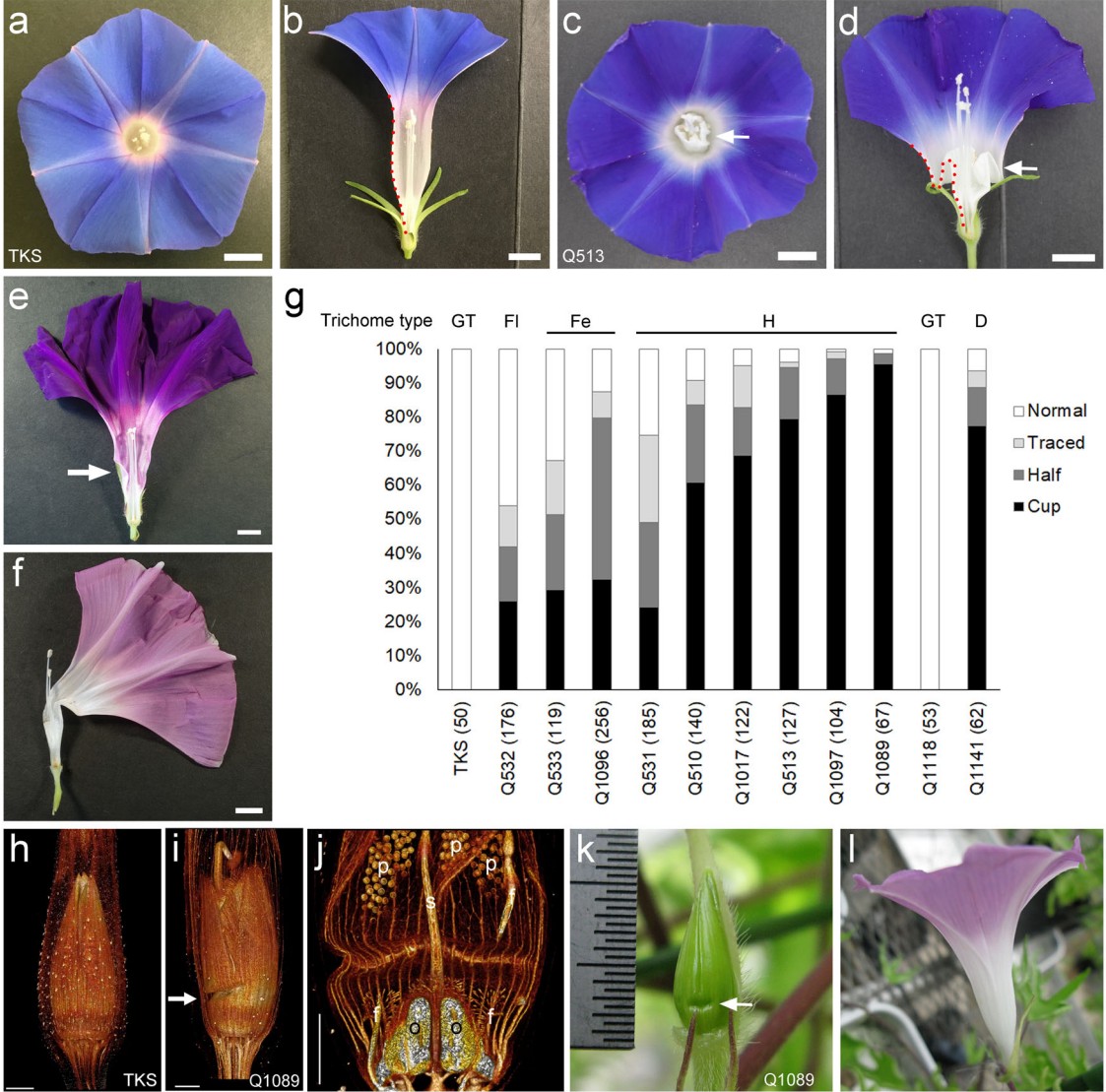

**Fig. 1 Corolla folding of cup-shaped flowers of morning glory. a, b** Top (**a**) and side (**b**) views of Tokyo Kokei Standard (TKS) flowers. **c, d** Top (**c**) and side (**d**) views of cup flower line Q513. The arrows indicate the cylinder-like structures generated by corolla folding. The tube shape is outlined with a red dotted line. **e** 'Traced' tube in Q533. **f** 'Half' tube in Q1089. **g** Ratio of corolla defects in cup flower lines. The examined flower number is shown with the line ID. Types of glandular secretory trichomes on the corolla are shown above the graph (see Fig. 3). GT normal glandular trichome, Fl flat, Fe few, H hair, and D deformed. **h–j** 3D constructed image of floral buds of TKS and Q1089 from X-ray microcomputed tomography imaging. **h** TKS. **i, j** Q1089. The arrow in (i) indicates the cavity. f filament, o ovule, p pollen, s style. **k, l** Sepal removal at the early stage restored the straight elongation of the corolla. The arrow in **k** indicates the cavity on the corolla surface. Scale bars: **a–f** 1 cm: **h–j** 1 mm.

higher von Mises stress at the buckling region and at the lower edge of the frictional area, meaning that the residual stress during development accumulated in these areas (Fig. 2c and Supplementary Movie 4). The residual stress structure is consistent with the actual morphology determined from the raw image data (Fig. 2d). The buckling shape was mimicked by a simple experiment with a thin elastic sheet to which force was applied from both sides (Fig. 2e). The simulated growth-induced buckling mimicked corolla folding in cup flower lines, suggesting that mechanical stresses from both sides, *i.e.*, acropetal growth or constraint by corolla attachment to the receptacle, and basipetal mechanical stress by physical friction, were sufficient to cause corolla folding.

**Defects in glandular secretory trichomes in cup flower mutants.** To investigate the cellular mechanism of friction, the surfaces of perianth organs were examined using scanning electron

microscopy. The TKS corolla formed a number of glandular secretory trichomes (GSTs) with four head secretion cells, similar to the type VI GST in *Lycopersicon* species (Fig. 3a, b)[21,22]. In Q513, almost no GSTs had developed on the corolla surface, and nonsecretory hair trichomes developed at the distal part of the corolla (Fig. 3c, d). We defined this type as the 'hair trichome' (H) type, which included Q510, Q513, Q531, Q1017, Q1089, and Q1097 (Supplementary Table 1). The Q532 corolla did not develop either GSTs or hair trichomes (Fig. 3e, f); thus, it was defined as 'flat' (Fl). The other lines produced fewer and smaller GST-like structures (Fig. 3g, h) and were defined as the 'few' (Fe) type, which included Q533 and Q1096. The cup flower line Q1141 of *I. purpurea* developed GST-like granules whose morphology was aberrant compared to those of the wild type and thus were defined as 'deformed' (D) (Fig. 3i, j). When the corolla was optically cleared, the GSTs of wild-type line Q1118 presented a brown color, whereas those of Q1141 did not (Fig. 3k, l), suggesting that the

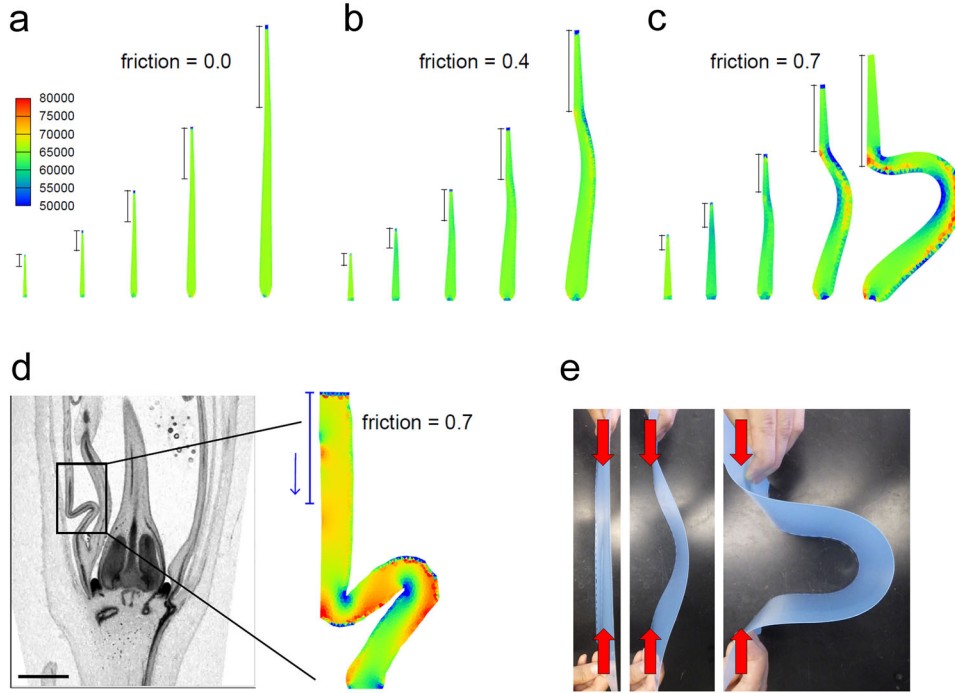

**Fig. 2 Computational model of corolla folding in morning glory. a–c** von Mises stress distribution during simulation without friction (**a**), with friction 0.4 (**b**), and with friction 0.7 (**c**). The bars indicate the frictional areas. The friction ranges from 0 to 1, where 0 means no inhibition of growth and 1 means no growth at the restricted area. **d** Estimated von Mises stress distribution with friction 0.7 based on the raw cross-sectional image of the corolla folding in the off-central region, with the same color code as that in (**a**). **e** Simple experiment with a thin polypropylene sheet showing mechanical buckling when force is applied from both sides.

deformed GSTs of Q1141 lack the secretion ability. GSTs were also produced on the adaxial side of sepals in TKS and clustered together more than those on the corolla did (Supplementary Fig. 5a, b). For the cup flower mutants, the GSTs on sepals were more scarce or were deformed (Supplementary Fig. 5c–h), indicating that GST development was defective on both the corolla and sepals. Compared with the flat lines, the hair-type lines showed a higher ratio of corolla folding (Fig. 1g), suggesting that hair trichomes enhanced corolla folding, thus supporting the mechanical stress model.

We hypothesized that secretion from GSTs reduces the friction between the corolla and sepals and examined the surface lipid content on the corolla and floral buds using thin-layer chromatography. We detected alkanes, esters, ketones, aldehydes, alcohols, and fatty acids but did not find significant differences between the TKS and cup flower mutants (Supplementary Fig. 6). As GSTs synthesize and accumulate large amounts of secondary metabolites, including terpenes, phenylpropanoid derivatives, acyl sugars, methyl ketones, and proteins, as well as fatty acid derivatives, in plants[23–25], one or a mixture of these metabolites could act as a lubricant for corolla elongation.

Another possible cause of corolla folding is that the corolla in the cup flower mutant is less rigid than that in TKS, *i.e.*, the flexural rigidity (FR) of corolla in the mutant is less than that in TKS. However, this might not be the case because the corolla in the mutant was thicker with more cell layers than that in TKS, and the cell wall thickness was not significantly different between the mutant and TKS (Supplementary Fig. 7). These suggest that the elasticity and the second moment of area, the components of FR, of corolla in the mutant is not significantly less than that in TKS. Therefore, we concluded that it is unlikely for the corolla in the cup flower mutant to be less rigid than that in TKS.

**Transcriptome of the cup flower mutant**. To investigate the molecular mechanism of corolla elongation, we examined transcripts in the corolla using RNA sequencing. We selected the Q532 'flat' line for corolla RNA extraction to determine the fundamental function of GSTs without hair-type trichomes. The reads were mapped to the *I. nil* genome[14], and we found that 1491 genes were selectively expressed in the corolla compared to the leaves (Fig. 4a). Self-organizing map (SOM) clustering showed that 477 genes were corolla specific and that their expression was downregulated in mutant corollas (Fig. 4b). Among the downregulated genes in the mutant corolla, 101 genes showed almost no expression in Q532 (Supplementary Data 1). We focused on transcription factors and checked their expression in TKS and cup flower lines: the results showed that *InMYB113* (INIL05g09649) was not detected in the floral buds of any cup flower line (Fig. 4c). *InMYB113* was classified as part of cluster 9 in the SOM analysis (Fig. 4b), suggesting that it is a corolla-specific gene. *InMYB113* is a putative ortholog of *AtMYB113* (At1g66370) and *AtMYB114* (At1g66380), belonging to subgroup 6 of the MYB gene family[26–29]. *InbHLH35* (INIL13g09091) showed different expression patterns in the cup flower lines (Fig. 4c) and is a putative ortholog of *AtbHLH35* (At5g57150), which belongs to subgroup IIIC, and its biological function is unknown[30,31]. MYB and bHLH genes are known to regulate epidermal cell fate, including GSTs, hair trichomes, and root hairs[27–29,32,33], suggesting that *InMYB113* and *bHLH35* regulate GST development in the perianth organs of morning glory.

## Discussion

Using morning glory mutants, we found that GSTs are involved in corolla elongation within floral buds. Since GSTs biosynthesize and accumulate secondary metabolites, we suggest that secretions from GSTs on perianth organs act as a lubricant and reduce the friction of floral organs, enabling the corolla to elongate straight

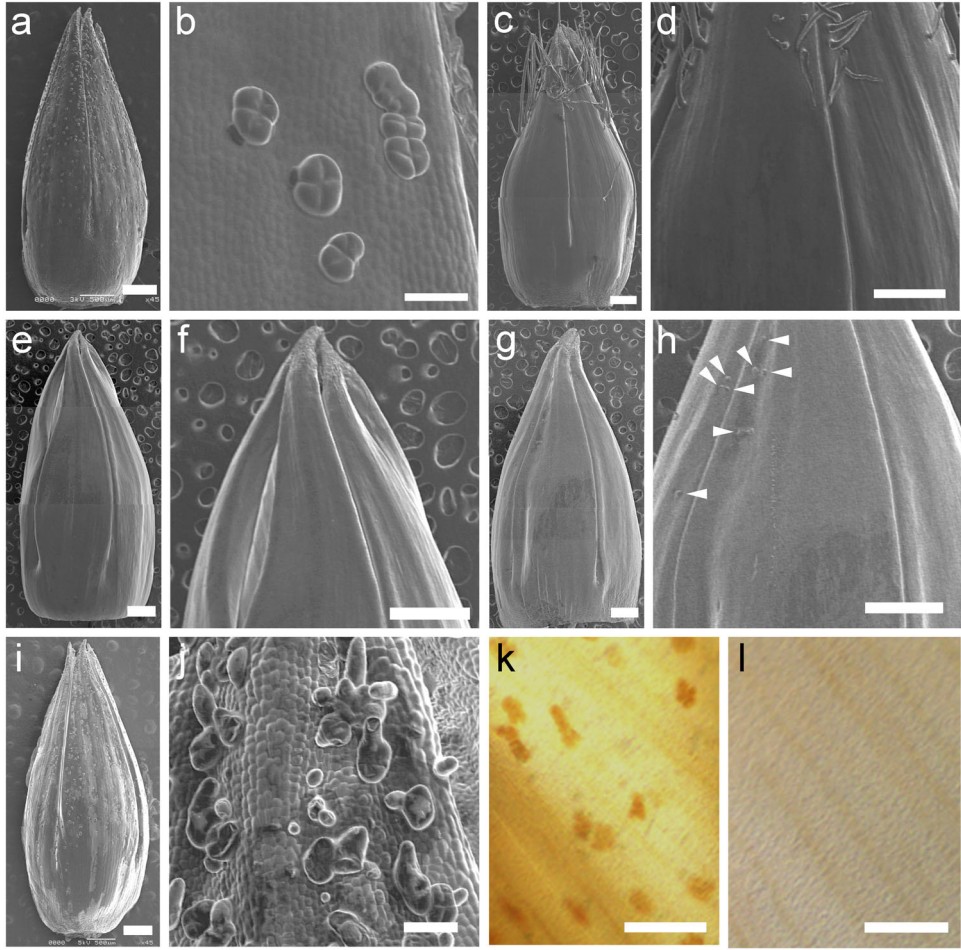

**Fig. 3 Glandular secretory trichomes on the abaxial side of a morning glory corolla. a, b** Tokyo Kokei Standard (TKS) corolla with glandular trichomes. **c, d** Q513 corolla without glandular trichomes. Note that hairy trichomes develop at the distal part of the corolla. **e, f** Q532 corolla without glandular trichomes. **g, h** Q1096 corolla with tiny structures (arrowheads). **i, j** Q1141 corolla with deformed glandular trichomes. **k, l** Corolla of Q1118 (**k**) and Q1141 (**l**) treated with clearing solution, showing that glandular trichomes are densely stained in Q1118 but that no such structures are present in Q1141. Scale bars: **a**, **c**, **e**, **g**, **i**, 500 μm; **b**, **d**, **f**, **h**, **j**, 50 μm; **k**, **l**, 100 μm.

properly (Supplementary Fig. 8). This suggests a novel function for GSTs in organogenesis, *i.e.*, mechanical regulation to reduce the conflicting stresses between surrounding organs, in addition to acting as mechanical and chemical barriers for plant defense[21,22,34]. Genetic and physiological analyses of GSTs constitute the next step to understand the molecular function of GSTs in corolla morphogenesis.

Previously, we examined two folded-petal mutants of *Arabidopsis thaliana* and found that wax/diacylglycerol synthesis and transport in petal cells were involved in straight elongation of petals within floral buds[35,36], suggesting that our lubricant model is applicable to both choripetalous plants (those with separated petals) and sympetalous plants (those with fused petals). In addition, GSTs in many plants are initiated at a very early stage on leaf primordia and develop by imbricating with the other primordia within leaf buds[37]; thus, GSTs may also reduce the friction of leaf primordia during early development. Since several plant species form GSTs on the tube part of the corolla, it is possible to develop novel valuable ornamental flowers with folded corollas by enhancing the friction between peripheral organs. One way involves the genetic ablation of GSTs in peripheral organs. Several promoters upstream of trichome genes in tobacco, tomato, cotton, and *Artemisia annua* have been shown to induce reporter gene expression in GSTs[38–42]; thus, these promoters could be used for the genetic ablation of GSTs.

Our findings propose a unique mechanical system in plant organogenesis that differs from elastic animal tissue development. Animal tissues such as those involved in gastrulation and vertebrate gut formation develop as a consequence of mechanical conflict[43]. Compared with these animal tissues, the corolla does not enhance mechanical conflict but rather actively reduces it to achieve proper organ development. As plant organs can suffer mechanical buckling through frictional stress, especially in complex structures such as flowers, they must maintain proper frictional conditions between organs spatiotemporally. Therefore, our results emphasize the importance of GSTs in tuning spatiotemporal friction during flower development, and thus, they are critical for the interplay between microstructures and macroscopic morphogenesis in plants.

## Methods

**Plant materials and growth conditions.** Ten lines of *I. nil* (L.) Roth (syn. *Pharbitis nil* (L.) Choisy) and two lines of *I. purpurea* (L.) Roth were provided by Kyushu University, Japan (Supplementary Table 1). Tokyo Kokei Standard (TKS; Q1065) and Q1118 were used as wild type for *I. nil* and *I. purpurea*, respectively. The *cp* lines included Q510, Q513, Q531, Q1017, Q1089, and Q1097, and the *cp-r* lines included Q532, Q533, and Q1096 for *I. nil*. The cup flower line Q1141 of *I. purpurea* was developed from Q1181 (which has a normal corolla), probably due to transposon mutations. Seeds were sown in garden soil (Nihon Hiryo Co., Ltd., Tokyo, Japan) and grown in a glass greenhouse under natural sunlight conditions or in a growth chamber under short days (8-h light and 16-h dark) or long days (16-h light and 8-h dark) at 25 to 28 °C.

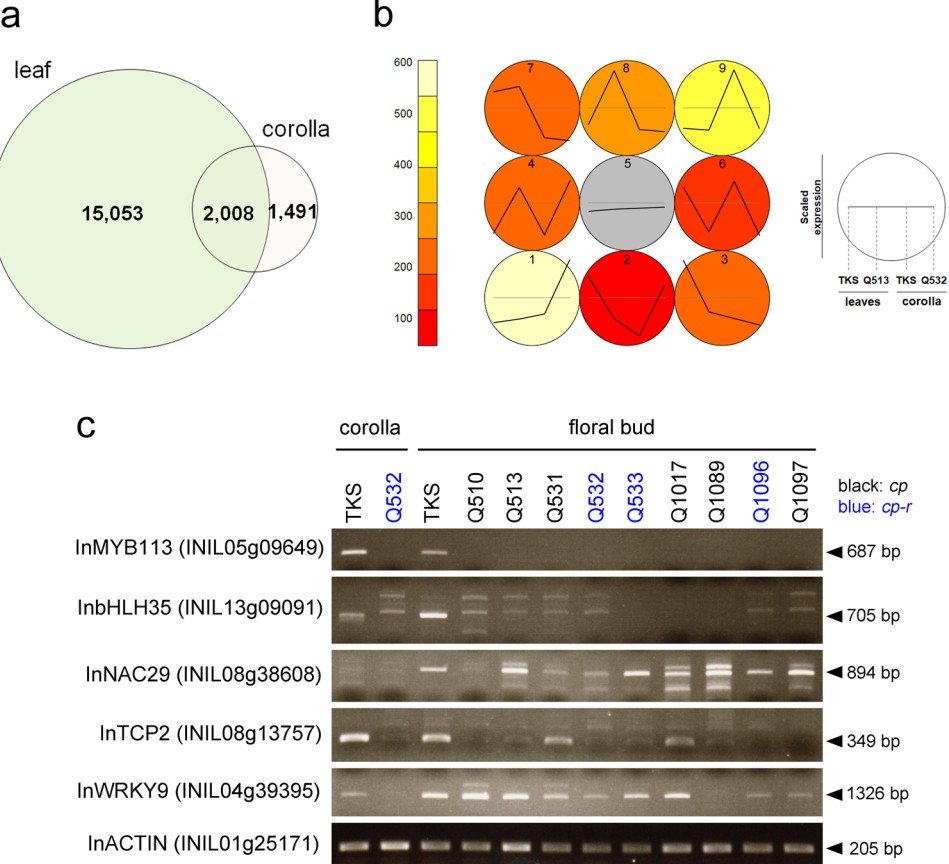

**Fig. 4 Expression analysis in morning glory cup flower mutants. a** Number of expressed genes in the leaves and corolla. **b** SOM clustering of expressed genes. The cluster number is shown in each circle. **c** RT-PCR results of transcription factor-encoding genes whose expression levels were downregulated in Q532, as revealed through RNA sequencing. The black and blue letters of the cup flower lines indicate *cp* and *cp-r*, respectively. The sample tissues were corolla tissues from TKS and Q532 and floral buds from TKS and cup flower lines. The expected band sizes are indicated.

**Microscopy**. Flower images were captured with a GX200 digital camera (Ricoh, Japan) and a S8AP0 stereomicroscope equipped with an EC3 device (Leica, Germany). For the morphology of glandular trichomes, floral buds at the early stage (the petal height was 5 to 7 mm) were examined with a JSM 5800LV scanning electron microscope (JEOL, Japan). For tissue clearing, floral buds were fixed in fixing solution (90% ethanol and 10% acetic acid, v/v) for 1 h, immersed in an ethanol series (90%, 80%, 70%, 60%, 50%, and 30%), rinsed with distilled water, and then immersed in clearing solution (20 g of chloral hydrate, 5 mL of glycerol, and 2.5 mL of distilled water) overnight. For paraffin sectioning, floral buds were embedded in Paraplast Plus (Sigma-Aldrich, Japan) after FAA (3.7% formaldehyde, 5% acetic acid, and 50% ethanol; v/v) fixation and an ethanol/lemosol series treatment. Sections that were 10 µm thick were prepared by a rotary microtome (Yamato Kohki Industrial Co. Ltd., Japan), and after deparaffinization with lemosol and subsequent ethanol series, they were stained in 0.05% toluidine blue (w/v) for 10 to 15 s. The cleared tissue and sections were investigated with a 2500 DM microscope equipped with a DFC450 C camera (Leica, Germany).

**X-ray micro-CT imaging and histology**. Floral buds of TKS and Q1089 were fixed in FAA solution, and after subsequent immersion in an ethanol series, the samples were soaked in contrast agent, 0.3% phosphotungstic acid in 70% ethanol solution for 77 days, or a 1:3 mixture of Lugol's solution and deionized distilled water for 2 days, as previously described[44–47]. The samples were then scanned using X-ray micro-CT (ScanXmate-E090S105, Comscantechno Co., Ltd., Kanagawa, Japan) at a tube voltage peak of 85 kV and a tube current of 90 µA. The samples were rotated 360° in steps of 0.24°, generating 1500 projection images of 992 × 992 pixels. For wide range scanning of the specimens at relatively high resolution, the specimens were scanned in four (TKS) or two (Q1089) parts per specimen. For 2D analysis, the micro-CT data were reconstructed at an isotropic resolution of 3.8 × 3.8 × 3.8 µm (TKS) or 6.0 × 6.0 × 6.0 µm (Q1089). For 3D analysis, the micro-CT data were reconstructed at an isotropic resolution of 6.3 × 6.3 × 6.3 µm (TKS) or 10.0 × 10.0 × 10.0 µm (Q1089). Three-dimensional tomographic images were obtained using the OsiriX software program (https://www.osirix-viewer.com/).

**Computational model of a growing corolla**. We constructed a continuous mechanical model simulating a growing petal with or without the friction of the outer organ to evaluated its effect on the resulting petal shapes. To do this, we implemented a model for sepal morphogenesis[48–50]. In this study, we considered only the vertical cross section of the petal as a two-dimensional elastic material. The petal shape at rest (step *n*) is inflated by turgor pressure *P*, reaching mechanical equilibrium with a certain boundary condition; afterward, it is used as a resting shape for the next step ($n + 1$). We used the boundary condition of a frictional area with reduced turgor pressure, thereby reducing growth due to friction with the outer organ. We also set the displacement of the horizontal axis as fixed, which corresponds to no horizontal movement of the petal by attaching to the outer organ. The combination of elastic growth and stress release corresponds to the viscous relaxation of the petal, similar to that of sepals[49,51]. The model was simulated in FreeFem + +[52]. We used generalized Hooke's law in conjunction with the stress tensor $\sigma$ and the strain tensor $\varepsilon$. The components of the stress tensor along with the principal and secondary stiffness directions ($\sigma_{11}, \sigma_{22}, \sigma_{12} = \sigma_{21}$) are related to those of the strain tensor by

$$\begin{pmatrix} \sigma_{11} \\ \sigma_{22} \\ \sigma_{12} \end{pmatrix} = \frac{E}{1-\nu^2} \begin{pmatrix} 1+\alpha & \nu\sqrt{(1+\alpha)(1-\alpha)} & 0 \\ \nu\sqrt{(1+\alpha)(1-\alpha)} & 1-\alpha & 0 \\ 0 & 0 & \frac{\beta}{1+\nu} \end{pmatrix} \begin{pmatrix} \varepsilon_{11} \\ \varepsilon_{22} \\ \varepsilon_{12} \end{pmatrix}$$

where the geometrical mean of Young's modulus *E* satisfies the following relationships: $E_1 = E(1 - \alpha)$ along the principal stiffness direction and $E_2 = E(1 + \alpha)$ along the secondary stiffness direction. The indices $\alpha$ and $\beta$ are the anisotropic growth parameters ($\alpha = 0$ in the isotropic case), and $\nu$ is the Poisson ratio. In this study, we used $\alpha = 0.1$, $\beta = 0.9$, $P = 4 \times 10^4$ (Pa), $E = 3 \times 10^6$ (Pa), and $\nu = 0.4$. The friction parameter $\eta$ is introduced with the reduced turgor pressure $P_{\text{red}} = (1 - \eta)P$. As an initial setup at step 0, the petal cross section is shaped as a rectangle with a width of 0.1 (mm) and a height of 0.8 (mm). After 200 steps, it deforms to reach its final shape resulting from the effect of friction. The Mises stress for each triangle element is calculated as $\text{MS} = \sqrt{\frac{(\sigma_{11}-\sigma_{22})^2}{2}}$ for each triangle. For the actual morphology of the petal cross section, we first extracted the petal

cross-sectional outline by the ImageJ selection tool and implemented it as an initial shape. As expected, there was residual stress resulting from the developmental process, and we applied a downward force with magnitude $4P$ that corresponds to the roughly estimated residual stress before reaching the shape in the image (blue arrows in Fig. 2).

**Surface lipid analysis of floral organs.** Floral buds from plants grown under 12 h of light and darkness at 25 °C were used. Surface lipids of corolla and calyx tissues were extracted by soaking them in chloroform twice for 30 s each and then concentrated under a nitrogen stream at 30 °C. Total lipids were extracted according to the Bligh–Dyer method[53] from a tissue sample that was flash frozen in liquid nitrogen and pulverized by beating with zirconia beads. Lipid classes were separated by thin-layer chromatography (TLC) on a silica gel 60 plate ($25 \times 25$ cm, Merck), with a hexane:ether:acetic acid (90:7.5:1, v/v) solvent system. Lipid spots were stained with spraying primuline solution (0.01% (w/v) in 80% acetone) and imaged under UV light. Surface lipids extracted from *Arabidopsis thaliana* inflorescence stems were used as standards.

**RNA sequencing and RT-PCR.** Total RNA was extracted from three independent corollas from TKS or Q532 flowers. Approximately 0.05 g of corolla tissue was used for RNA extraction according to a modified protocol with an RNeasy Plant Mini Kit[54] (QIAGEN, Germany). After the RNA integrity was confirmed by running samples on an Agilent RNA 6000 Nano Chip (Agilent Technologies, USA), 0.5 µg of the total RNA samples was used for library preparation for RNA-seq analysis. Libraries were prepared using an Illumina TruSeq Stranded mRNA LT Sample Kit according to the manufacturer's instructions (Illumina, USA). The pooled libraries were subsequently sequenced on an Illumina NextSeq 500 sequencing platform, and single-end reads that were 76 bp long were obtained. The count data were then subjected to trimmed mean of M-value (TMM) normalization in EdgeR. Transcript expression and digital gene expression (DEG) were defined using the edgeR GLM approach. The reads were mapped to the coding sequence data of *I. nil* by using BWA[55], and 44,916 genes were obtained. The count data were then subjected to trimmed mean of M-value (TMM) normalization in edgeR[56], after which transcript expression and digital gene expression (DEG) were defined using the edgeR GLM approach[56]. We selected 845 and 904 genes whose expression was downregulated 904 upregulated, respectively, by summing those with a false discovery rate (FDR) < 0.01, a sum (total number of mapped reads) > 1, and a $\log_2FC < -1$ (downregulated) or $\log_2FC > 1$ (upregulated). For Gene Ontology (GO) analysis, we used the PANTHER classification system through TAIR (https://www.arabidopsis.org/) database[57]. For RT-PCR, cDNA synthesis was performed using ReverTra Ace qPCR RT Master Mix together with a gDNA remover (Toyobo, Japan). cDNA from the corolla of TKS and Q532 and from the floral buds of TKS, Q510, Q513, Q531, Q532, Q533, Q1017, Q1089, Q1098, and Q1097 was examined. Two microliters of template DNA, 0.1 µL of TaKaRa ExTaq DNA polymerase, 2 µL of 10x ExTaq Buffer, 1 µL of a 2.5 mM dNTP mix, 1 µL of 10 µM forward and reverse primers (Supplementary Table 2), and 12.9 µL of sterilized distilled water were combined to prepare a 20 µL reaction solution for PCR, and the amplified product was confirmed via 1% agarose gel electrophoresis.

**Reporting summary.** Further information on research design is available in the Nature Research Reporting Summary linked to this article.

## Data availability
RNA sequencing data have been deposited in the DBJJ database under accession number DRA010590. Source data for Fig. 1g is shown in Supplementary Data 2. Any remaining information can be obtained from the corresponding author upon reasonable request.

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

## Acknowledgements
We thank A. Hoshino (NIBB), H. Fujita (NIBB), M. Ono (Tsukuba University), and S. Sakaguchi (Nara Women's University) for helpful suggestions for this research and the National Bio-Resource Project of the AMED, Japan, for the plant materials. This work was supported by JSPS KAKENHI JP18K06366 to S. Takeda, MEXT KAKENHI JP18H05484 to S. Tsugawa and T.D., MEXT KAKENHI JP18H0548 to T.D., JSPS KAKENHI JP18H04787 and MEXT Grant Number S1511023 to S.K., and NIG-JOINT (44A2020) to S. Takeda.

## Author contributions
S. Takeda and K.O. conceived and designed the study. S.Takeda, A.S., S.Ts., K.O., M.T., A.M., T.S., K.S., N.T., and M.N. performed the experiments and analyzed the data. E.N. provided the plant materials, S. Tsugawa performed the computational modeling, and A.S., S. Takeda, and S. Tsugawa drafted the manuscript, with critical revisions from E.N., T.D., and K.O. All the authors have approved the final version of the manuscript.

## Competing interests
The authors declare no competing interests.
