## [Peer Review File · Communications Biology]

Reviewers' comments:

Reviewer #1 (Remarks to the Author):

The manuscript entitled "Reduction of organ-organ friction is critical for corolla elongation in morning glory" by Shimoki et al. focused on organ-organ interaction during corolla development, which provides an interesting field for plant morphogenesis research. In this manuscript, the authors highlighted the importance of GSTs (glandular secreting trichomes) in regulating the organ-organ interactions for corolla morphogenesis. However, I still some comments as below:

1. The background part is quite short, and limited information was introduced about the floral organ development and nothing about "morning glory" was mentioned. Also, current progress of plant corolla elongation should be introduced in depth.
2. Line 60 – 61 of Page 3. The authors declared "Crosses between TKS, cp, and cp-r suggest that corolla folding is a recessive trait and is caused by the same gene in cp and cp-r", It's not clear to me why corolla folding is caused by the same gene in cp and cp-r. Besides, as the F1 leaves of cp and cp-r are semi-wrinkled noted in supplementary Fig. 2, which is insufficient to reach the conclusion that corolla folding is a recessive trait.
3. In Abstract part, the authors declared that "defects in GST development in perianth organs results in folding of the corolla tube". However, in corresponding Result section, I only found that GST were associated with corolla folding, but did not find that the folding of corolla tube was caused by GST.
4. In the Discussion part, the authors should discuss their findings in depth, not just retell the story from the Results section.

Reviewer #2 (Remarks to the Author):

The manuscript by Shimoki, Tsugawa and colleagues proposes that the reduction in inter-organ friction is an important factor shaping flower morphology in morning glory. Authors first describe in details the morphology of the cp mutant flowers and suggest that bulking of their corolla is caused by increased friction between corolla and sepals. They test this hypothesis in computer simulation that convincingly matches the biological observations. The bulking of corolla is proposed to result from the defects in glandular secretory trichomes development which are reduced in the mutant flowers. Finally, by RNAseq they find candidate genes that could potentially underlie the observed trichomeless phenotype and suggest that InMYB113 may be the gene responsible for this phenotype.

This study is very interesting and proposes a novel mechanism of the control of floral organ morphogenesis. The idea of glandular trichomes being involved in reduction the frictions is novel and could well be the reason for the observed phenotypes. However, this is not the only possibility. Apart of having impaired trichome development, mutant plants have clear growth phenotype visible for example in the leaves. It could be that the bulking of the corolla is caused by modified, differential growth of the sepals and corolla. Reduced growth in sepals or increased growth in corolla could lead to the corolla bulking as there may be less space for it to elongate. Authors do not consider this possibility. In my opinion, this is the major problem with the conclusions of this manuscript. Could the bulking be produced in simulations if instead of reduced friction, the models would simulate a reduced space for the development of the corolla? Authors should for example precisely measure the size of

different floral organs at critical stages of the mutant flower development and compare it to the wild type. Without additional experiments excluding this possibility, the current conclusion of the manuscript (reduced organ-organ friction) may not be correct.

Reviewer #3 (Remarks to the Author):

This is a very interesting paper that describes the effects of mechanical constraints on organ shape. It shows how friction between sepals and the corolla of morning glory lead to inward buckling and formation of an altered shape in a mutant that likely affects a MYB transcription factor. The key demonstration is that removal of the sepals leads to straightening of the corolla in the mutant. The authors further show that glandular secretory trichomes are reduced in the mutant and propose that this causes increased friction between sepals and corolla. However, the authors do not consider an alternative hypothesis that the mutant affects the rigidity of the corolla tube. It is possible that in wild type the corolla tube is sufficiently rigid to overcome the frictional effect of the sepals but in the mutant the corolla tissue is flimsier and thus buckles. A structural defect of this kind might also explain the leaf phenotype. The authors need to fully discuss and test this alternative hypothesis before they can claim that reduced friction is involved. In particular, they should determine the thickness of the developing corolla cell walls (particularly the epidermal walls) in the mutant compared to wild type (e.g. by transmission electron microscopy). Mechanical tests such as stretching the corolla or leaves to breaking point would also confirm that there are no changes in mechanical properties of the corolla. Also, the authors' hypothesis predicts that trichomes should be fully developed at points of sepal-petal contact in wild type at the developmental stage when buckling is first evident in the mutant. Is this true? If so, it should be shown.

I would like to emphasise that should the authors show that the reason for the buckling is a change in mechanical properties of the corolla, not reduced friction, the paper would still be of great interest as it shows the effect of interactions between corolla and sepal. What matters is to evaluate and test both hypotheses rigorously.

I also have some detailed/presentation comments:

1. I find it confusing when the authors describe mechanical stresses coming from both directions. What would stress coming from one direction mean? As I understand it the authors are proposing that mechanical constraints operate on both sides of the corolla tube. At the base there is constraint from attachment to the receptacle and other organs, while higher up there is the constraint of friction against the sepals. Removing one of these constraints (sepal removal) is sufficient to prevent residual stress accumulating.
2. Fig.1k describes the effect of sepal removal yet a cavity is visible on the corolla surface – why?
3. What happens if only one sepal is removed? Does the cavity only disappear in the region next to it? Does the corolla extend more on one side?
4. It would help to show images of sepals at early stages in relation to the corolla. Are they attached to each other to form a constraining ring around the corolla? Where are the points of closest contact with the corolla of wild type? This is relevant to point 3 above.

Reviewer #1 (Remarks to the Author):

The manuscript entitled "Reduction of organ-organ friction is critical for corolla elongation in morning glory" by Shimoki et al. focused on organ-organ interaction during corolla development, which provides an interesting field for plant morphogenesis research. In this manuscript, the authors highlighted the importance of GSTs (glandular secreting trichomes) in regulating the organ-organ interactions for corolla morphogenesis. However, I still have some comments as below:

1. The background part is quite short, and limited information was introduced about the floral organ development and nothing about "morning glory" was mentioned. Also, current progress of plant corolla elongation should be introduced in depth.

> Thank you for this constructive comment. In the revised introduction section, we reorganized it with general interest on mechanical regulation in organ morphogenesis, the functional role and growth properties of corolla, wide variation and database of morning glory, and construction of this manuscript.

2. Line 60 – 61 of Page 3. The authors declared "Crosses between TKS, cp, and cp-r suggest that corolla folding is a recessive trait and is caused by the same gene in cp and cp-r", It's not clear to me why corolla folding is caused by the same gene in cp and cp-r. Besides, as the F1 leaves of cp and cp-r are semi-wrinkled noted in supplementary Fig. 2, which is insufficient to reach the conclusion that corolla folding is a recessive trait.

> To explain this part more clearly, we added data to the Supplementary Fig 3. First, cp and cp-r are caused by the same gene because F1 plants generated the cup flower (Supplementary Fig. 3c). Second, cp is the recessive because cp and TKS cross results in normal flower in F1 generation, and subsequent F2 generation the normal flower and cup flower segregated to 3:1 ratio. We added these results to the main text (line 107-116).

3. In Abstract part, the authors declared that "defects in GST development in perianth organs results in folding of the corolla tube". However, in corresponding Result section, I only found that GST were associated with corolla folding, but did not find that the folding of corolla tube was caused by GST.

> Thank you for this important comment. Since the corolla attached to the sepal and anther in the floral buds, there is a space only in the tube part to fold inward. We added the section photos to Supplementary Fig. 7 to show this clearly.

4. In the Discussion part, the authors should discuss their findings in depth, not just retell the story from the Results section.

> We added Discussion section to support the importance of our findings. In the revised discussion section, in addition to the importance of future analysis of GSTs, we added a possible contribution to horticulture study and uniqueness of our simulation study in terms of growth-induced buckling.

Reviewer #2 (Remarks to the Author):

The manuscript by Shimoki, Tsugawa and colleagues proposes that the reduction in inter-organ friction is an important factor shaping flower morphology in morning glory. Authors first describe in details the morphology of the cp mutant flowers and suggest that bulking of their corolla is caused by increased friction between corolla and sepals. They test this hypothesis in computer simulation that convincingly matches the biological observations. The bulking of corolla is proposed to result from the defects in glandular secretory trichomes development which are reduced in the mutant flowers. Finally, by RNAseq they find candidate genes that could potentially underlie the observed trichomeless phenotype and suggest that InMYB113 may be the gene responsible for this phenotype.

This study is very interesting and proposes a novel mechanism of the control of floral organ morphogenesis. The idea of glandular trichomes being involved in reduction the frictions is novel and could well be the reason for the observed phenotypes. However, this is not the only possibility. Apart of having impaired trichome development, mutant plants have clear growth phenotype visible for example in the leaves. It could be that the bulking of the corolla is caused by modified, differential growth of the sepals and corolla. Reduced growth in sepals or increased growth in corolla could lead to the corolla bulking as there may be less space for it to elongate. Authors do not consider this possibility. In my opinion, this is the major problem with the conclusions of this manuscript. Could the bulking be produced in simulations if instead of reduced friction, the models would simulate a reduced space for the development of the corolla? Authors should for example precisely measure the size of different floral organs at critical stages of the mutant flower development and compare it to the wild type. Without additional experiments excluding this possibility, the current conclusion of the manuscript (reduced organ-organ friction) may not be correct.

> Thank you for the critical comments. First, the leaf phenotype and corolla folding are supposed to be caused by different genes, since cp-r lines bear cup flower but normal leaves. We added this data to the Supplementary Fig. 3a. Second, we checked the histological and micro CT images and confirmed that growth of sepals and corolla was not different significantly between TKS and cup flower mutants. The corolla length was measured (added to Supplementary Fig. 4), showing little difference of corolla size. These suggest that there is little or no difference for space in the floral buds. We added the section photos to Supplementary Fig. 7 and main text (line 185-191). Since all cup flower lines showed defects in GSTs development, we think we can conclude that GSTs defects are the major cause for corolla folding.

Reviewer #3 (Remarks to the Author):

This is a very interesting paper that describes the effects of mechanical constraints on organ shape. It shows how friction between sepals and the corolla of morning glory lead to inward buckling and formation of an altered shape in a mutant that likely affects a MYB transcription factor. The key demonstration is that removal of the sepals leads to straightening of the corolla in the mutant. The authors further show that glandular secretory trichomes are reduced in the mutant and propose that this causes increased friction between sepals and corolla. However, the authors do not consider an alternative hypothesis that the mutant affects the rigidity of the corolla tube. It is possible that in wild type the corolla tube is sufficiently rigid to overcome the frictional effect of the sepals but in the mutant the corolla tissue is flimsier and thus buckles. A structural defect of this kind might also explain the leaf phenotype. The authors need to fully discuss and test this alternative hypothesis before they can claim that reduced friction is involved. In particular, they should determine the thickness of the developing corolla cell walls (particularly the epidermal walls) in the mutant

compared to wild type (e.g. by transmission electron microscopy). Mechanical tests such as stretching the corolla or leaves to breaking point would also confirm that there are no changes in mechanical properties of the corolla. Also, the authors' hypothesis predicts that trichomes should be fully developed at points of sepal-petal contact in wild type at the developmental stage when buckling is first evident in the mutant. Is this true? If so, it should be shown.

I would like to emphasise that should the authors show that the reason for the buckling is a change in mechanical properties of the corolla, not reduced friction, the paper would still be of great interest as it shows the effect of interactions between corolla and sepal. What matters is to evaluate and test both hypotheses rigorously.

>We appreciate these fruitful comments very much. To examine the corolla structure (and thus we can speculate the rigidity of corolla), we checked the histology and micro CT sections, and found that cup flower lines had thicker corolla than TKS. The cell structure did not seem to be so different, and more cell layers were confirmed on cup flower corolla than that of TKS (Supplementary Fig. 7). As the flexural rigidity is determined by the elastic property (Young's modulus E) and the cross-sectional shape property (moment of inertia of area I), it is theoretically expected that cup flower corolla has a higher I than wild type because I is proportional to h^3 with the thickness h . As we observed the cup flower have thicker morphology, we would say that the flexural rigidity that causes the mechanical buckling is still high level even if the cup flower becomes more or less soft (lower Young's modulus). Therefore, we suppose that the rigidity of corolla may not be the dominant factor of the corolla folding. For the stage of GSTs development, they fully grow at the corresponding stage of concavity occurs in the mutant, as shown in Fig. 3 (SEM) images.

I also have some detailed/presentation comments:

1. I find it confusing when the authors describe mechanical stresses coming from both directions. What would stress coming from one direction mean? As I understand it the authors are proposing that mechanical constraints operate on both sides of the corolla tube. At the base there is constraint from attachment to the receptacle and other organs, while higher up there is the constraint of friction against the sepals. Removing one of these constraints (sepal removal) is sufficient to prevent residual stress accumulating.

>We appreciate this comment. The buckling occurs when the mechanical constraint comes from both directions, as pointed out. We re-think our model figure (Supplementary Fig. 8), and mechanical stress from below actually comes from either growth or attachment to the receptacle. This may be indirect stress due to the one from above, but still give constraint to the buckling point of corolla. According to this idea, we changed the words in the figure from 'growth' to 'growth or constraint from attachment to the receptacle'.

2. Fig.1k describes the effect of sepal removal yet a cavity is visible on the corolla surface – why?

>This is because the concavity had been already produced when we removed the sepal. Even so, corolla elongate straight afterwards, so that the stress release after concavity formation still restored the corolla elongation. This supports our idea that the stress within the restricted space (floral buds) caused the corolla buckling.

3. What happens if only one sepal is removed? Does the cavity only disappear in the region next to it? Does

the corolla extend more on one side?

4. It would help to show images of sepals at early stages in relation to the corolla. Are they attached to each other to form a constraining ring around the corolla? Where are the points of closest contact with the corolla of wild type? This is relevant to point 3 above.

>We added the section images of floral buds to Supplementary Fig. 7, showing that the sepals surround the corolla tightly (but each sepal is not fused each other), and that corolla and sepals contact at the lower (tube) part. It is almost impossible to see the concavity from outside of sepals, and there are variation of corolla folding phenotype as shown in Fig. 1g, so we did not examine the one-sepal removal experiment.

Reviewers' comments:

Reviewer #1 (Remarks to the Author):

The authors have well addressed my comments.

Reviewer #2 (Remarks to the Author):

Thanks for addressing my concerns.

Reviewer #3 (Remarks to the Author):

The authors have gone part way to addressing my comments and have now mentioned the alternative hypothesis of altered corolla strength in the Results. I remain concerned, however, that they have still not rigorously ruled out the alternative explanation. Indeed, their new result that the mutant corolla is thicker suggests an alteration in corolla mechanical properties. I note that reviewer 2 also raises the same issue of an alternative hypothesis so this needs to be properly addressed before the paper can be accepted.

I suggest the following should be done:

1. As I understand it, panels k and l of supp fig 7 indicate no change in corolla thickness in the mutant but the other panels do suggest a change in thickness. This discrepancy needs to be resolved.
2. The internal structure of the thicker corolla in the mutant needs to be clarified. Is it thicker because the cells are larger/more spaced out or because the cell walls are thicker? If they are more spaced out, then rigidity may be less because a smaller proportion of the cross-section is cell wall. Cross sections should be taken at the appropriate level and fraction of the corolla occupied by cell wall quantified.
3. A developmental time course needs to be given for the secretory glands to show that they are mature in wild type at the first stage at which buckling occurs in the mutant. I cannot tell from Fig.3b whether the trichome morphology shown is the fully differentiated form, what the stage of bud development is relative to buckling initiation, or the region in the corolla that the trichomes derive from. Staging also needs to be clarified for Fig. 3k and l to show that the darker staining arises in wild type at the time when buckling first occurs in the mutant.
4. Given the observed change in corolla thickness, the authors should ideally mechanically stretch the corolla to show that it has unaltered properties (e.g. Young's modulus). Extensometers for plant tissues have been developed in several labs.

REVIEWERS' COMMENTS:

Reviewer #3 (Remarks to the Author):

The authors have clarified the points I raised, and I am happy to recommend acceptance subject to the following modifications:

In response to point 1, error bars and results from significance tests need to be shown and given for Supp Fig. 7 to clarify which differences are significant. I also suggest the authors label the panels so that the identity of sepals and petals is clear.

In response to point 2, I welcome the clarification that the petal thickness increase is due to cell number. The authors explain this difference as being a result of genetic background or a secondary effect of mechanical stress. However, it is also possible that corolla is weaker or altered as a result of the mutation, and this leads to increased growth in thickness. Without measurements on cell wall thickness (not just petal thickness) or tensile strength the alternative hypothesis of a weaker corolla cannot be ruled out. I would strongly urge the authors to be more careful in their claims and give due weight to alternative possibilities throughout.

In response to point 3, the text/legends need to be modified to ensure these points about staging are clear to the reader.

In response to point 4, I accept that this experiment is difficult but without it the authors cannot rule out the alternative hypothesis of altered corolla properties and need to state this clearly in their text.

Point-to-point response to reviewer's comments (Manuscript ID: COMMSBIO-20-2627A)

Reviewer #3 (Remarks to the Author):

The authors have gone part way to addressing my comments and have now mentioned the alternative hypothesis of altered corolla strength in the Results. I remain concerned, however, that they have still not rigorously ruled out the alternative explanation. Indeed, their new result that the mutant corolla is thicker suggests an alteration in corolla mechanical properties. I note that reviewer 2 also raises the same issue of an alternative hypothesis so this needs to be properly addressed before the paper can be accepted.

I suggest the following should be done:

1. As I understand it, panels k and l of supp fig 7 indicate no change in corolla thickness in the mutant but the other panels do suggest a change in thickness. This discrepancy needs to be resolved.

>We quantified the corolla thickness in micro-CT images in Supp Fig 7, and found that the mutant corolla was thicker than that of TKS. We added this data to the graph, together with one from histological sections, in Supp Fig 7.

2. The internal structure of the thicker corolla in the mutant needs to be clarified. Is it thicker because the cells are larger/more spaced out or because the cell walls are thicker? If they are more spaced out, then rigidity may be less because a smaller proportion of the cross-section is cell wall. Cross sections should be taken at the appropriate level and fraction of the corolla occupied by cell wall quantified.

>We counted the cell number of the corolla layer on the histological sections, and found that TKS corolla had around 7 cell layers, whereas the *cp* mutants had 8~9 cell layers (Supp Fig 7). As we can see in the cross sections, the cells of the *cp* mutants do not space out more than TKS corolla, i.e., the cell sizes are comparable (Supp Fig 7). Theses suggest that thicker corolla in *cp* mutants is due to increased cell number to the radial direction, but not to the change of cell size. Moreover, as we explained in the previous response, it is theoretically expected that the flexural rigidity that causes the mechanical buckling is still high level (due to moment of inertia I including thickness) even if the cup flower becomes more or less soft (lower Young's modulus). Therefore, we suppose that the flexural rigidity of corolla (a product of material elasticity and moment of inertia) may not be the dominant factor of the corolla folding. We suppose that the increased cell number in corolla is due to either genetic background or secondary effect from mechanical stress, as shown in the text (line 188-191). We believe that our results are enough to conclude that the mutant corolla is not less rigid than TKS, which is sufficient for our mechanical model.

3. A developmental time course needs to be given for the secretory glands to show that they are mature in wild type at the first stage at which buckling occurs in the mutant. I cannot tell from Fig.3b whether the trichome morphology shown is the fully differentiated form, what the stage of bud development is relative to buckling initiation, or the region in the corolla that the trichomes derive from. Staging also needs to be clarified for Fig. 3k and l to show that the darker staining arises in wild type at the time when buckling first occurs in the mutant.

>The buckling starts when corolla hight reaches to around 8 mm, as shown in Supp Fig 4_b1. The stage of the SEM analysis in Fig 3 is when corolla hight reaches to 4 mm. At this earlier stage the glandular trichomes fully developed on TKS corolla (Fig 3a and b: their morphology is not different from GSTs on later stage corolla.). Also, the mutant panels on Fig 3 are almost at the same stage (corolla hight around

4 mm), and showed no buckling yet, suggesting that they are at the stage before buckling starts. Therefore, we can conclude that GSTs are fully developed before corolla buckling initiate. Fig 3k and l are the same stage with the Fig 3 SEM panels.

4. Given the observed change in corolla thickness, the authors should ideally mechanically stretch the corolla to show that it has unaltered properties (e.g. Young's modulus). Extensometers for plant tissues have been developed in several labs.

>Thank you for this constructive suggestion. We know that it would be great if we could measure the strength of corolla directly with extensometers, but this is technically impossible, since corolla is very soft and thin, and easy to break, and it is difficult to extract small intact corolla from young floral buds without affecting its structure. Therefore, instead of direct measurement we performed mathematical modelling (Fig 2), and showed the normal corolla extension when we removed calyx at early stage in the mutant (Fig 1).